# Practice Recommendations for the Management of MASLD in Primary Care: Consensus Results

**DOI:** 10.3390/diseases12080180

**Published:** 2024-08-10

**Authors:** Christos Lionis, Sophia Papadakis, Marilena Anastasaki, Eftihis Aligizakis, Foteini Anastasiou, Sven Francque, Irini Gergianaki, Juan Manuel Mendive, Maria Marketou, Jean Muris, Spilios Manolakopoulos, Georgios Papatheodoridis, Dimitrios Samonakis, Emmanouil Symvoulakis, Ioanna Tsiligianni

**Affiliations:** 1Clinic of Social and Family Medicine, School of Medicine, University of Crete, 71003 Heraklion, Greece; sophiapapadakis@gmail.com (S.P.); iriniger@hotmail.com (I.G.); esymvoulakis@uoc.gr (E.S.); i.tsiligianni@uoc.gr (I.T.); 2Health Center of Kandanos, Crete, 73004 Kandanos, Greece; aligizak@gmail.com; 34th Local Health Team-Municipality Practice and Academic Unit of Heraklion, Crete, 71303 Heraklion, Greece; fanast@hotmail.com; 4Department of Gastroenterology and Hepatology, Antwerp University Hospital, University of Antwerp, 2650 Edegem, Belgium; sven.francque@uza.be; 5La Mina Primary Health Care Centre, IDIAP Jordi Gol, 08003 Barcelona, Spain; juanmmendive@gmail.com; 6European Society for Primary Care Gastroenterology, London E1 6HU, UK; 7Clinic of Cardiology, University Hospital of Heraklion, Crete, 70013 Heraklion, Greece; maryemarke@yahoo.gr; 8Department of Family Medicine, Care and Public Health Research Institute (CAPHRI), Maastricht University, 6229 Maastricht, The Netherlands; jean.muris@maastrichtuniversity.nl; 9Department of Gastroenterology, National and Kapodistrian University of Athens, 11527 Athens, Greece; smanolak@med.uoa.gr; 10Gastroenterology Department, Medical School, National and Kapodistrian University of Athens, General Hospital of Athens “Laiko”, 11527 Athens, Greece; gepapath@med.uoa.gr; 11Clinic of Gastroenterology & Hepatology, University Hospital of Heraklion, 70013 Heraklion, Greece; dsamonakis@gmail.com

**Keywords:** non-alcoholic fatty liver disease, metabolic dysfunction-associated steatotic liver disease, primary care, consensus, Europe, recommendations

## Abstract

Background: Despite its high prevalence and impact on health, metabolic dysfunction-associated steatotic liver disease (MASLD) is inadequately addressed in European primary care (PC), with a large proportion of cases going undiagnosed or diagnosed too late. A multi-country European research consortium led a project to design and evaluate a patient-centered, integrated model for MASLD screening, diagnosis, and linkage to specialty care for European PC settings. Based on the lessons from this project, the latest research evidence, and existing guidelines for the management of MASLD, we sought to develop a set of practice recommendations for screening, referral, and management of MASLD in PC. Methods: The Rand/UCLA modified Delphi panel method, with two rounds, was used to reach consensus on practice recommendations. The international panel consisted of experts from six countries, representing family medicine, gastroenterology, hepatology, cardiology, and public health. Initially, fifteen statements were drafted based on a synthesis of evidence from the literature and earlier findings from our consortium. Prior to the consensus meeting, the statements were rated by the experts in the first round. Then, in a hybrid meeting, the experts discussed findings from round one, adjusted the statements, and reassessed the updated recommendations in a second round. Results: In round one, there was already a high level of consensus on 10 out of 15 statements. After round 2, there were fourteen statements with a high degree of agreement (>90%). One statement was not endorsed. The approved recommendations addressed the following practice areas: risk screening and diagnosis, management of MASLD–lifestyle interventions, pharmacological treatment of MASLD/MASH, pharmacological treatment for co-morbidity, integrated care, surgical management, and other referrals to specialists. Conclusions: The final set of 14 recommendations focuses on increasing comprehensive care for MASLD in PC. The recommendations provide practical evidence-based guidance tailored to PC practitioners. We expect that these recommendations will contribute to the ongoing discussion on systematic approaches to tackling MASLD and supporting European PC providers by integrating the latest evidence into practice.

## 1. Introduction

Chronic liver disease is a major cause of morbidity and mortality in Europe and worldwide. Metabolic dysfunction-associated steatotic liver disease (MASLD), formally called non-alcoholic fatty liver disease (NAFLD), and, in particular, metabolic dysfunction-associated steatohepatitis (MASH), a subtype of MASLD, are major causes of chronic liver disease [1,2,3,4,5,6,7]. The prevalence of MASLD is over 25% in European adults, with a significant increase in the incidence of MASLD/MASH of 1% per year worldwide [7,8]. Rates of MASLD among patients with obesity and/or type 2 diabetes mellitus (T2DM) have been shown to exceed 70% [2,8]. Importantly, MASLD is associated with disturbed cardiometabolic function and increased incidence of cardiovascular diseases (CVDs), dyslipidemia, insulin resistance, T2DM, and hypertension, which are components of the metabolic syndrome [1,3,9,10,11,12,13,14,15]. It is increasingly recognized that general practitioners (GPs) and other primary care providers (PCPs) can play a crucial role in the prevention, early detection (case finding), and long-term management of the MASLD spectrum [1,3,16,17]. Despite the existence of clinical guidelines and various efforts in some European settings, MASLD and MASH have so far received insufficient attention from PCPs in Europe and internationally, with a large proportion of cases going undiagnosed or diagnosed too late [1,3,7,18,19]. 

Advanced fibrosis is part of the spectrum of MASLD predominantly associated with chronic liver disease and its detection in patients with obesity, T2DM, and dyslipidemia, and is therefore particularly important; however, its diagnosis is not yet standard practice in European primary care [7,19]. There are limited studies that have examined physician knowledge, attitudes and practice patterns, and barriers related to MASLD/MASH [18,19,20,21,22]. Poor familiarity with MASLD/MASH guidelines, low confidence in the knowledge and skills necessary for addressing MALD/MASH, and the complexity of the disease have been identified as barriers to patient care by PCPs [18,19,20,21,22,23]. The available data indicate that most clinicians have not read clinical practice guidelines or received continuing medical education concerning MASLD/MASH [18,19,22,23]. Although European clinical practice guidelines have been published, they do not specifically outline the clinical role of PCPs compared to specialists, nor the key tools and referral and co-management practices that PCPs should be using as part of the co-management of disease risk and progression. There are calls for more involvement and education of primary care physicians and other health care professionals on MASLD/MASH guidelines [1,3,21,22,23]. A recent report by the European Association for the Study of the Liver (EASL)–*Lancet* Liver Commission highlighted that training for PCPs should particularly focus on health promotion and the prevention of liver disease, and the diagnosis of this disease at an earlier stage, indicating that primary care is particularly suited to addressing these areas of clinical practice [1]. Francque et al. (2021) also provided important guidance regarding the educational material that PCPs should use when meeting their patients [24]. 

To address this challenge, a Εuropean team of hepatology specialists, general practitioners, nurses, cardiologists, and public health specialists sought to consolidate the available evidence and provide expert advice on a concise set of practice recommendations for primary care.

## 2. Materials and Methods

The Rand/UCLA modified Delphi panel method, with two rounds, was used to reach consensus on pre-defined statements [25].

### 2.1. Composition of Expert Panel

The international panel consisted of fifteen experts from four countries (Greece, Belgium, The Netherlands, and Spain), representing family medicine, gastroenterology, hepatology, cardiology, and public health (Appendix A). The panel included selected experts in the field of MASLD, the researchers from three academic institutions participating in a European collaborative project, which aimed to design and evaluate a patient-centered, integrated model for MASLD screening, diagnosis, and linkage to specialty care within the European context [26], and selected academics from the University of Crete. The consensus meeting marked the final stage of this European project.

### 2.2. Consensus Process

Initially, fifteen statements were drafted based on a synthesis of evidence made by the core group of the University of Crete [CL, SP, MA] from the literature and the findings of this project. Prior to the consensus meeting, these statements were circulated and rated by the experts in the first round. As part of the first round, panelists were asked to indicate the extent to which they agreed with each statement. The response options included: strong disagreement (1), high disagreement (2), neutral (3), high agreement (4), and strong agreement (5). In the first round, the panel members could comment on the ranking they had indicated. The questionnaire, which contained 15 statements, appears in Appendix A. Consensus was defined as an agreement of seventy-five percent or more among the panel members. Subsequently, during an eight-hour hybrid meeting, the experts discussed the findings of the first round, adjusted the statements, and rescored the updated recommendations in a second round until a consensus was reached.

## 3. Results

### 3.1. First Round Results

Twelve experts ranked the statements in round one. The three panel members who prepared the recommendations did not participate in the round one ranking. Table 1 presents the results of the rankings from rounds one and two. Ten statements received initial consensus as part of round one. 

### 3.2. Consensus Meeting and Round Two Rankings

As part of the consensus meeting, a series of short presentations were provided to update the panel members on the latest evidence and key findings of the ‘MASLD Models of Care’ project. This was followed by a presentation of the round one rankings with a detailed summary of the comments received from the expert panel members for each statement. This was followed by a discussion of each recommendation. 

Following the consensus panel discussion, each statement was reviewed individually, and changes were made to the statements based on the consensus panel agreement. Members of the consensus panel then completed a second round in which they ranked their consensus with the revised statements. Table 1 provides the round two consensus panel rankings. During the consensus process, there was a high level of agreement on 14 statements (>90%). One statement (statement R13) was not endorsed by the panelists due to a lack of supporting evidence. 

A short description of the panel discussion is provided below.

(A)Risk Screening and Diagnosis

**R1**: General practitioners (GPs) and other primary health care (PHC) professionals should consider persons with indications of metabolic dysfunction, including overweight or obesity, type 2 diabetes, determinants of metabolic syndrome, and/or persistently elevated aminotransferase levels as ‘high risk’ for developing MASLD and MASH.

**Comments:** 
*There has been much discussion about the definition of metabolic dysfunction and its potential overlap with the determinants of metabolic syndrome. Those with evidence of metabolic dysfunction were identified as ‘high risk’ for developing MASLD and MASH. The panel acknowledges that terminology should be modified to properly identify individuals at increased risk of developing MASLD. Both obesity and type 2 diabetes are believed to have the greatest impact on the development of MASLD.*


**R2**: GPs and other PHC professionals should use calculators to predict liver fibrosis and, primarily, the FIB-4 index, which has been shown to be readily accepted and implemented in primary care settings.

**Comments:** 
*Non-invasive testing has been discussed as a method of determining the degree of fibrosis and the term, “non-invasive testing”, has been suggested to replace the term “prediction calculators”. The latter term may lead to confusion about the type and nature of prediction calculators. The panel determined that the Fibrosis 4 score (FIB-4) should be performed to assess the degree of fibrosis.*


**R3**: Transient elastography is recommended as the appropriate imaging technique to assess the degree of fibrosis in people with MASLD. 

**Comments:** 
*In a case where fibrosis is suspected, the original statement stipulates that transient elastography should be used. The recommendation of the panel was to replace transient elastography with a liver stiffness measurement (LSM, elastography) as a more general term not limited to a specific device. LSM cannot precisely stage fibrosis; as it has a high NPV, it is helpful in ruling out significant/advanced fibrosis or cirrhosis (depending on the threshold used). At present, this is mostly performed in tertiary centers. However, there could be programs with portable machines to screen the population at risk after selection by a GP. The recommendation should reflect that there will be local variation in access to and settings in which elastography is performed.*


**R4**: GPs and other PHC professionals should use prediction calculations to assess cardiovascular (CVD) risk, including the CVD score by the European Society of Cardiology.

**Comments:** 
*It has been underlined that people with MASLD have a higher risk of CVD and it has been agreed that management should be based on a treatment plan informed by risk assessment and laboratory values. This is already usual practice in most European settings. There may be other national risk calculators in use, and this should be acknowledged.*


**R5**: Persons with a mild or moderate risk of advanced liver disease should be assessed with FIB-4 test every 6 months.

**Comments:** 
*Non-invasive tests are well suited (although poorly validated) to monitoring the progression of MASLD and the timing of recurrence was discussed by the panel. Every six months proved too soon after the initial assessment for patients at moderate risk, and probably 12 to 24 months is an appropriate frequency. For patients at indeterminate risk with factors that persist, further assessment may be warranted sooner. There should be a thorough initial evaluation and, if the patient is diagnosed with advanced fibrosis, referral to a specialist is recommended with follow-up as deemed appropriate by a GP and specialist. Ambiguous results require further confirmation of the score with other tests, even invasive ones in some cases.*


(B)Referral to Specialists

**R6**: Persons with a high risk of advanced disease (FIB-4 ≥ 2.67 and/or abnormal transient elastography as above 7.9 kPa) should be referred to a specialist for further assessment and treatment.

**Comments:** 
*It was agreed that it was valuable to provide a cut-off for liver stiffness. Transient elastography has been replaced by the LSM. It was also agreed that referral may depend on the vitality, age, and personal preference of patients. It was discussed that, after the initial evaluation by the specialist, follow-up could be carried out by a primary care physician and should include screening for hepatocellular carcinoma (HCC) and biochemical determination of alpha-foetoprotein (aFP) at least two times per year.*


(C)Management of MSALD–Lifestyle Interventions and Risk Factor Management

**R7**: Persons with a high risk of advanced disease (FIB-4 ≥ 2.67 and/or abnormal transient elastography, above 7.9 kPa) should be supported for weight reduction and smoking cessation in primary care and referred to specialized services if needed.

**Comments:** 
*Transient elastography has been replaced by liver stiffness measures (LSMs). It was agreed that GPs and PHC practitioners are the most appropriate health care providers to look at the plan for lifestyle intervention and engage other specialists (e.g., dieticians, smoking cessation clinics) in supporting the patient’s treatment.*


**R8**: GPs and other PHC professionals should offer interventions including weight loss, smoking cessation, and restrictions on alcohol use for the management of MASLD/MASH. 

**Comments:** 
*It was agreed that the recommendation should be revised to highlight the importance of controlling metabolic risk factors. It was also agreed that the statement should be revised to ensure that the interventions offered are evidence-based.*


**R9**: GPs and other PHC professionals should use very brief advice and motivational interviewing interventions for lifestyle change in every consultation with a patient with a high-risk of or a confirmed diagnosis of MASLD/MASH.

**Comments:** 
*It was noted that motivational interviewing techniques may be useful; however, the evidence base is not clear enough for a strong recommendation to be made. The feasibility of having primary care play an important role in lifestyle interventions at every consultation was discussed and it was agreed this would be adapted.*


**R10**: GPs and other PHC professionals should promote MASLD/MASH awareness and health literacy among their patients.

**Comments:** 
*This statement should be revised from ‘should promote’ to ‘should be actively raising’. It was agreed that the term “health literacy” will not be clear to everyone and should be deleted. Finally, this recommendation was revised to focus on ‘patients at increased MASLD risk’ rather than ‘all patients’ as initially stated. It was agreed that printed educational materials would ideally be provided.*


(D)Pharmacological Treatment for MASLD/MASH

**R11**: In patients with MASLD and biopsy-proven MASLD and T2DM, GPs and other PHC professionals should consider treatment with GLP-1 RAs and pioglitazone.

**Comments:** 
*Incretin hormone agonists and, in particular, Glucagon-like protein1 receptor agonists (GLP1 RAs), with beneficial effects on obesity and on cardiovascular and renal outcomes, attracted the panelists’ attention, as has, to the same extent, another antidiabetic drug, pioglitazone, which has shown an effect on steatohepatitis. As the evidence is still evolving, the panel decided to recommend that GPs remain in consultation with specialists and make a decision based on the latest evidence.*


(E)Pharmacological Treatment For Co-morbidity

**R12**: To reduce the cardiovascular risk in patients with MASLD/MASH, GPs and other PHC professionals should consider treatment with GLP1 RAs, pioglitazone, or SGLT2 inhibitors.

**Comments:** 
*As mentioned above, the panel decided to recommend that GPs stay in consultation with specialists and make a decision based on local guidelines.*


**R13** **:** GPs and other PHC professionals should consider semaglutide 2.4 mg/week or liraglutide 3 mg/day as a treatment option for individuals with MASLD or MASH and a BMI => 27 kg/m^2^ as adjunctive therapy to promote lifestyle modification and improve cardiovascular risk.

**Comments:** 
*This statement is not supported and has been omitted from the recommendations. The panel accepted that, although there is some evidence that GLP-1 agonists can improve histological lesions of the liver, additional evidence is needed to document safety and efficacy in patients with advanced liver disease.*


(F)Surgical Management

**R14**: GPs and other PHC professionals should consider bariatric surgery as a therapy along with improvement of the cardiovascular risk in persons with MASLD and a BMI of 35 kg/m^2^ or more (in the European population) and refer them to a specialist for a final decision.

**Comments:** 
*The panel concluded that although bariatric surgery shows beneficial and lasting effects in terms of weight loss, its effectiveness and the long-term safety of various techniques in patients with MASLD remain questionable. For this reason, the panel decided to modify the original underlying statement to “appropriate referral may be considered based on local referral criteria and patient preferences”.*


(G)Integrated Care

**R15**: GPs and other PHC professionals should collaborate with laboratory personnel and specialists to promote the health and well-being of patients with MASLD.

**Comments:** 
*The panel decided to replace the recommendation to collaborate with laboratory personnel and specialties along with interaction with a multidisciplinary care team. The recommendation to promote the health and well-being of patients with MASLD was supplemented by the word “management”.*


Final recommendations

The approved recommendations are provided in Box 1 and are illustrated in Figure 1.

Box 1Consensus Statements: Practice Recommendations for Screening, Referral, and Management of MASLD in European Primary Care.(A)Risk Screening and Diagnosis**1.** General practitioners (GPs) and other primary health care (PHC) professionals should consider persons with indications of metabolic dysfunction, including overweight or obesity, type 2 diabetes, and/or persistently elevated liver enzymes as at increased risk for developing MASLD. **2.** GPs and other PHC professionals should use non-invasive tests to estimate the risk of liver fibrosis and particularly the FIB-4 index, which has been shown to be easily accepted and implemented in primary care settings.**3.** Persons at intermediate or high risk of fibrosis based on first-line assessment require further investigation of liver stiffness (elastography), according to local pathways.**4.** All persons at increased risk of MASLD according to R1 should have a CVD risk assessment, based on prediction tools such as the CVD score by the European Society of Cardiology.**5.** Persons with a low or intermediate risk of advanced fibrosis should be assessed with FIB-4 in primary care periodically.(B)Referral to specialists**6.** Persons at high risk of fibrosis (FIB-4 ≥ 2.67 and/or abnormal liver stiffness tests above 7.9 kPa) despite lifestyle changes, should be referred to a liver specialist for further assessment and treatment.(C)Management of MSALD–Lifestyle Interventions and Risk Factor Modification**7.** Persons at high risk of fibrosis (FIB-4 ≥ 2.67 and/or abnormal liver stiffness tests above 7.9 kPa) should be supported for weight reduction and/or smoking cessation in primary care and referred to weight management and/or smoking cessation services as needed.**8.** GPs and other PHC professionals should offer effective and person-centered interventions including weight loss, smoking cessation, and restrictions on alcohol use for the management of MASLD. **9.** GPs and other PHC professionals should routinely offer effective lifestyle change support that may include very brief advice or motivational interviewing interventions in patients with a high risk or confirmed diagnosis of people with MASLD.**10.** GPs and other PHC professionals should actively raise MASLD awareness among persons at increased MASLD risk.(D)Pharmacological Treatment for MASLD/MASH**11.** In patients with biopsy-proven MASH and T2DM, GPs in consultation with specialists could consider treatment with medication which may include GLP1 RAs and/or pioglitazone with appropriate assessment and follow-up. (E)Pharmacological Treatment For Co-morbidity**12.** In patients with MASLD, reducing CVD risk should be prioritized, and GPs could consider consultation with other specialists and treatment with medication in accordance with local guidelines.(F)Surgical Management**13.** Evidence supports the benefits of bariatric surgery on CVD and fibrosis risk for patients with MASLD/MASH and obesity, and appropriate referral could be considered according to local referral criteria and patient preference.(G)Integrated Care**14.** GPs and other PHC professionals should interact with a multidisciplinary care team, including community services, to promote the health, well-being, and management of patients with MASLD.

## 4. Discussion

### 4.1. Summary

There is an urgent need to support PCPs in Europe in the appropriate screening, referral, and management of MASLD in patients in their practices. Fourteen practice recommendations were developed across five domains that include: risk screening and diagnosis, the management of MASLD and lifestyle interventions, pharmacological treatment of MASLD/MASH, pharmacological treatment for co-morbidity, referral to specialists, surgical management, and integrated care. 

The recommendations developed by the consensus panel highlight the important role of PCPs in screening patients at high risk of MASLD with five practice recommendations. Specifically, the consensus panel identified the importance of risk screening among patients with an indication of metabolic dysfunction (overweight or obesity, T2DM) and/or persistently elevated liver enzymes. The role of non-invasive tests that assess the risk of liver fibrosis in primary care and, in particular, the FIB-4 index, has been highlighted. Risk assessment can then be used to guide additional assessment, with individuals at intermediate or high risk of fibrosis requiring further investigation of liver stiffness with elastography, recognizing that local pathways determine the setting in which elastography can take place. Elastography is performed in some contexts in primary care and others in specialty care in addition to CVD risk assessment. Persons at high risk for fibrosis (FIB-4 ≥ 2.67 and/or abnormal liver stiffness tests, above 7.9 kPa, should be referred to a liver specialist for further assessment and treatment. Periodic repeat FIB-4 assessment is recommended for individuals at low or intermediate risk of advanced fibrosis. 

A key recommendation of the practice guidance is the need for tight control of metabolic risk factors in persons at increased risk of MASLD alongside patient education about steatotic liver disease, its risk, and risk reduction. Practice recommendations identify the need for weight reduction and smoking cessation including referral to specialist services for persons at high risk of fibrosis defined as a FIB-4 ≥ 2.67 and/or abnormal liver stiffness tests. Furthermore, patients can be supported with lifestyle interventions, including the development of a treatment plan, and patient-centered counseling to support lifestyle change is recommended. For patients with increased MASLD risk and a diagnosis of MASLD, priority should be given to lowering CVD risk and the medical management of risk factors should be optimized through pharmacological treatment in consultation with specialists, recommending bariatric surgery in patients presenting with obesity with MASLD/MASH as an evidence-based treatment for lowering CVD and fibrosis risk. The practice recommendations also emphasize the importance of integrated care models between primary care, specialty care, and community-based service providers to effectively manage the risk of disease progression. 

### 4.2. Impact

These practice recommendations should not be seen as an attempt to develop a new set of guidelines or replace the existing and published guidance. Rather, they are designed to facilitate the daily practice of primary care physicians and offer prompt and evidence-based information to more clearly define the role of primary care in terms of specific aspects of the screening and management of MASLD/MASH. The practice recommendations highlight the significant role of primary care and assist with operationalizing these activities in busy primary care practices in Europe. They provide simple guidance regarding the specific role of PCPs and the key role of integrated care with specialist colleagues. The practice recommendations are also supported via an eLearning course produced by the team members and available via the ESPCG website (https://www.espcg.eu/nafld/ accessed on 6 August 2024).

Importantly, we have seen growing interest internationally in supporting PCPs in integrating MAFLD/MASH screening and management into clinical practice routines with the publication of clinical practice guidelines and recommendations for primary care. US-based guideline committees have published MAFLD practice guidance specifically for primary care [27,28]. However, the current consensus is based on actual research implemented in the primary care setting, and it merits attention [26]. As we have done with the development of the present practice recommendations, future updates to European MAFLD practice guidelines, an effort that is underway under the coordination of the European Association of Study Liver (EASL), may seek to identify practice guidance regarding the role of primary care alongside specialist colleagues and support their dissemination via training and other modalities to PCPs in Europe.

### 4.3. Strengths and Limitations

The main limitations stem from the Delphi process, which depends on expert opinion. The practice recommendations reflect the opinion of a multidisciplinary panel, and the ranking is limited to the opinions and expertise of the consensus panel. While the first round of the Delphi process included 12 participants, 4 participants left the meeting before it was concluded due to work-related responsibilities. As such, the second voting round was conducted with eight participants. The number of panelists may also be considered low and was missing representation from other important disciplines, particularly nutritionists, since the type of diet and the presence (or absence) of nutritional factors are crucial for achieving weight reduction and, importantly, maintaining the desired BMI as part of MASLD management. There could be concerns regarding representation because patients were not involved in the consensus process, nor were leading liver organizations such as the EASL. Nevertheless, one of the strongest points of this initiative lies in the fact that it is based on real data collected during the rollout of a pilot intervention aimed at evaluating a management algorithm developed through discussions with program participants and experts. 

## 5. Conclusions

Fourteen practice recommendations were drafted after an expert review of current evidence and approved by a multidisciplinary body of clinicians and researchers. The translation and dissemination of these practice recommendations for use in a wider European setting is expected to improve the management of MASLD in primary care. This consensus is expected to contribute to the future update of European guidelines for this condition. 

## Figures and Tables

**Figure 1 diseases-12-00180-f001:**
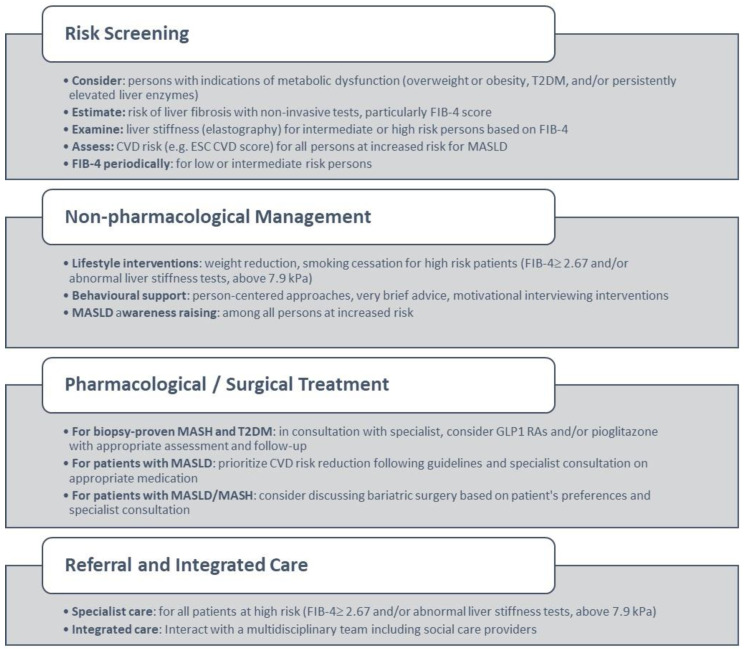
Primary care management of MASLD based on consensus results. Abbreviations: T2DM: Type 2 diabetes, CVD: cardiovascular disease, ESC: European Society of Cardiology.

**Table 1 diseases-12-00180-t001:** Round one and two consensus panel rankings.

Statement	Round 1		Round 2	
	N	% Agree	N	% Agree
**R1**	12	92	8	100
**R2**	12	92	8	100
**R3**	12	67	8	100
**R4**	12	100	8	100
**R5**	12	58	8	100
**R6**	12	92	8	100
**R7**	12	92	8	100
**R8**	12	92	8	100
**R9**	12	83	8	100
**R10**	12	75	8	100
**R11**	12	58	8	87.5
**R12**	12	58	8	87.5
**R13**	12	33	8	Rejected
**R14**	12	83	8	100
**R15**	12	75	8	100

## Data Availability

Data is maintained within this article.

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
