# Peer review of "Practice Recommendations for the Management of MASLD in Primary Care: Consensus Results"

_diseases, 2024, doi:10.3390/diseases12080180_

Round 1
Reviewer 1 Report
Comments and Suggestions for Authors
Expert are from 4 (greece, belgium, netherlands, spain) and not 6 countries.
The manuscript although concise is well-written and presented.
A figure-diagram summarizing the primary management would fix more the messages in addition to a list of recommendations.
Author Response
Thanks for your points and comments and please see our responses in the attached file.

Reviewer 2 Report
Comments and Suggestions for Authors
Great paper and a crucial one at this time. Agree that the Primary care doctor (PCP) is the first point of entry into the health care system for patients and as such have a great role in play in early identification and prevention of MASLD/MASH. PCPs also see patients with type 2 DM, obesity and metabolic syndrome often.
The methodology is appropriate however the number of participants appear to be too few. Agree with including this in the limitation section. For guidelines that would be applied to PCPs, may be great to have more PCPs on the panel. Also in the first round there were 12 panelists and 8 in the second round. Why did 4 panelists drop off?
Author Response
Dear Reviewer,
Many thanks for all your points and comments.
Comments and Suggestions for Authors
- Great paper and a crucial one at this time. Agree that the Primary care doctor (PCP) is the first point of entry into the health care system for patients and as such have a great role in play in early identification and prevention of MASLD/MASH. PCPs also see patients with type 2 DM, obesity and metabolic syndrome often.
Response: We thank the reviewer for this comment.
- The methodology is appropriate however the number of participants appear to be too few. Agree with including this in the limitation section. For guidelines that would be applied to PCPs, may be great to have more PCPs on the panel. Also in the first round there were 12 panelists and 8 in the second round. Why did 4 panelists drop off?
Response: We thank the reviewer. To address these concerns, we have added in the Limitations section the following clarifications: “While the first round of the Delphi process included 12 participants, four participants left the meeting before its end due to work-related responsibilities. As such, the second voting round was conducted with eight participants. The number of panelists may also be considered low and missing representation from other important disciplines, particularly nutritionists, since the type of diet and the presence (or absence) of nutritional factors are crucial for achieving weight reduction and, importantly, maintaining the desired BMI as part of MASLD management”.
Reviewer 3 Report
Comments and Suggestions for Authors
The study focuses on addressing metabolic dysfunction-associated steatotic liver disease (MASLD) in European primary care settings. The aim was to develop a set of practice recommendations for MASLD screening, referral, and management in primary care based on lessons from a multi-country European research project. Utilizing the Rand/UCLA modified Delphi panel method, a consensus was reached on fourteen statements through discussions and ranking by experts from various medical fields. The final recommendations aim to improve comprehensive care for MASLD by providing evidence-based guidance tailored to primary care practitioners.
In my opinion, the work is very interesting and highlights the necessity of unifying doctors' positions on this issue. However, I missed the voice of nutritionists. While BMI reduction is a positive factor in the course of the disease, the type of diet and the presence (or absence) of nutritional factors are crucial for maintaining the desired BMI. The role of nutrition should not be limited to weight reduction or pharmacological support. This concern does not detract from my positive assessment of this work but emphasizes the need for a more holistic approach.
Author Response
Comments and Suggestions for Authors
- The study focuses on addressing metabolic dysfunction-associated steatotic liver disease (MASLD) in European primary care settings. The aim was to develop a set of practice recommendations for MASLD screening, referral, and management in primary care based on lessons from a multi-country European research project. Utilizing the Rand/UCLA modified Delphi panel method, a consensus was reached on fourteen statements through discussions and ranking by experts from various medical fields. The final recommendations aim to improve comprehensive care for MASLD by providing evidence-based guidance tailored to primary care practitioners.
Response: We thank the reviewer for summarizing the scope and the results of this study.
- In my opinion, the work is very interesting and highlights the necessity of unifying doctors' positions on this issue. However, I missed the voice of nutritionists. While BMI reduction is a positive factor in the course of the disease, the type of diet and the presence (or absence) of nutritional factors are crucial for maintaining the desired BMI. The role of nutrition should not be limited to weight reduction or pharmacological support. This concern does not detract from my positive assessment of this work but emphasizes the need for a more holistic approach.
Response: We appreciate this comment. To acknowledge this important remark we have added the following text to our Limitations section: “The number of panelists may also be considered low and missing representation from other important disciplines, particularly nutritionists, since the type of diet and the presence (or absence) of nutritional factors are crucial for achieving weight reduction and, importantly, maintaining the desired BMI as part of MASLD management”
Round 2
Reviewer 1 Report
Comments and Suggestions for Authors
The manuscript has substantially improved.